UPDATE ARTICLE

# Inhibitory feedback from the motor circuit gates mechanosensory processing in *Caenorhabditis elegans*

**Sandeep Kumar**[1], **Anuj K. Sharma**[2], **Andrew Tran**[1], **Mochi Liu**[2], **Andrew M. Leifer**[1,2]*

**1** Princeton Neuroscience Institute, Princeton University, Princeton, New Jersey, United States of America,
**2** Department of Physics, Princeton University, Princeton, New Jersey, United States of America

* leifer@princeton.edu

The Editors encourage authors to publish research updates to this article type. Please follow the link in the citation below to view any related articles.

## Abstract

Animals must integrate sensory cues with their current behavioral context to generate a suitable response. How this integration occurs is poorly understood. Previously, we developed high-throughput methods to probe neural activity in populations of *Caenorhabditis elegans* and discovered that the animal's mechanosensory processing is rapidly modulated by the animal's locomotion. Specifically, we found that when the worm turns it suppresses its mechanosensory-evoked reversal response. Here, we report that *C. elegans* use inhibitory feedback from turning-associated neurons to provide this rapid modulation of mechanosensory processing. By performing high-throughput optogenetic perturbations triggered on behavior, we show that turning-associated neurons SAA, RIV, and/or SMB suppress mechanosensory-evoked reversals during turns. We find that activation of the gentle-touch mechanosensory neurons or of any of the interneurons AIZ, RIM, AIB, and AVE during a turn is less likely to evoke a reversal than activation during forward movement. Inhibiting neurons SAA, RIV, and SMB during a turn restores the likelihood with which mechanosensory activation evokes reversals. Separately, activation of premotor interneuron AVA evokes reversals regardless of whether the animal is turning or moving forward. We therefore propose that inhibitory signals from SAA, RIV, and/or SMB gate mechanosensory signals upstream of neuron AVA. We conclude that *C. elegans* rely on inhibitory feedback from the motor circuit to modulate its response to sensory stimuli on fast timescales. This need for motor signals in sensory processing may explain the ubiquity in many organisms of motor-related neural activity patterns seen across the brain, including in sensory processing areas.

## Introduction

A critical role of the nervous system is to detect sensory information and to select a suitable motor response, taking into consideration the animal's environment and current behavior. How the brain integrates sensory stimuli with broader context is an active area of research. For

**Data Availability Statement:** Computer-readable files showing processed tracked behavior trajectories and stimulus events for all experiments are publicly posted at https://doi.org/10.25452/figshare.plus.23903202. All analysis code used in this manuscript are available at https://github.com/leiferlab/kumar-sensorimotor-integration.git Transgenic strains AML17, AML496 and AML499 and plasmid RRID:Addgene_195853 are being made available through the Caenorhabditis Genetics Center (CGC) and Addgene respectively.

**Funding:** Research reported in this work was supported by the National Science Foundation (https://www.nsf.gov) through an NSF CAREER Award to AML (IOS-1845137) and through the Center for the Physics of Biological Function (PHY-1734030); and by the National Institute of Neurological Disorders and Stroke (https://www.ninds.nih.gov/) of the National Institutes of Health, National Institute of Neurological Disorder and Stroke under New Innovator award number DP2-NS116768 to AML; and by the Simons Foundation (https://www.simonsfoundation.org/) under award SCGB 543003 to AML. The funders had no role in study design, data collection and analysis, decision to publish, or preparation of the manuscript.

**Competing interests:** The authors have declared that no competing interests exist.

example, primates integrate a primary visual cue with a contextual visual cue to flexibly alter their neural computations [1,2]. In *Drosophila*, dopaminergic signals reflect mating drive, a long-lived internal state, that in turn gates the animal's courtship response to auditory and visual cues [3]. In *Caenorhabditis elegans* long-lived internal states lasting many minutes such as hunger [4], quiescence [5–9], and arousal [10] are all thought to alter the animal's response to stimuli via various synaptic or neuromodulatory mechanisms and have also been shown to alter the animal's mechanosensory response [11,12]. In those investigations, sensory signals are combined with one another or are integrated with long-lived internal state. Less is known about how sensory processing is modulated by short-timescale behavior. Short seconds-time-scale modulation of sensory processing is of particular interest because (1) it allows the animal to respond to urgent signals, such as threats; and (2) because the timescale suggests a circuit level mechanism, instead of other longer timescale mechanisms, such as neuromodulation or changes in gene expression. Here, we investigate short-timescale behavioral modulation of the *C. elegans* gentle-touch response.

We study the nematode *C. elegans* because its compact brain is well suited for investigations spanning sensory input to motor output [13,14]. The *C. elegans* gentle-touch circuitry allows the animal to avoid predation and is one of the most well-studied circuits of the worm [15–17]. We previously discovered that animals traveling forward are much more likely to respond to a mechanosensory stimulus by backing up (reversal), than animals that receive the same stimulus while they are in the middle of a turn [18,19]. In other words, the worm's response to mechanosensory stimuli is gated by the animal's short-timescale behavioral context. Suppressing mechanosensory-evoked reversals during turns may be part of a predator avoidance strategy. Turns are an important part of the *C. elegans* escape response, and by preventing turns from being interrupted prematurely, the animal may be ensuring that the escape response continues to completion [18,20,21].

The neural mechanism underlying this rapid modulation of sensorimotor processing has not previously been described. Because turns are short lived, lasting less than 2 s, we suspect gating is mediated by fast neural dynamics at the circuit level.

In mouse, fly and *C. elegans*, regions across the brain exhibit activity patterns related to the animal's locomotory state and body pose [22–25]. A leading hypothesis is that these motor signals may be important to modulate sensory representations including but not limited to vision [26], thermosensation [27], or corollary discharge [27–29]. In this study, we sought to investigate how locomotory signals interact on short timescales with downstream mechanosensory-related signals to modulate mechanosensory processing.

We previously developed a high-throughput closed-loop optogenetic approach [19] to interrogate the mechanosensorimotor circuitry in *C. elegans*. Here, we use this method to explore downstream mechanosensory processing by activating or inhibiting neurons associated with generating turns and reversals. We measure the animal's behavior in response to over 97,000 stimulus events. From these measurements, we identified a putative circuit by which inhibitory signals from turning-associated neurons disrupt mechanosensory processing and modulates the likelihood of a reversal depending on the animal's behavior.

## Results

### Turns on their own decrease the likelihood of mechanosensory-evoked reversals

Previously, we reported that optogenetic activation of all six gentle-touch mechanosensory neurons delivered during forward locomotion appeared more likely to evoke a transition to backward locomotion, called a "reversal," than activation delivered during the onset of a turn

[18]. We then developed high-throughput methods to probe this behavior with greater statistical power and concluded that either turning itself or possibly some other behavior related to turning modulates mechanosensory-evoked reversals (Fig 1A–1C, S1–S3 Videos) [19].

We sought to distinguish whether turns themselves modulated the reversals or whether it was another ancillary behavior related to turns. Turns in our previous recordings most often occurred immediately after backward locomotion—part of a fixed action pattern called the "escape response" that consists of backward locomotion, a turn and then finally forward locomotion [20]. By contrast, about 44% of the turns we observed were preceded by only forward locomotion, what we call "isolated" turns. We sought to test whether isolated turns also exhibited a reduction in mechanosensory-evoked responses.

By reanalyzing our prior measurements [19], we found that isolated turns also reduced the likelihood of a reversal response (Fig 1C and 1D). This finding suggests that turns alone are sufficient to modulate the likelihood of a mechanosensory-evoked reversal response. We therefore focused on the turn regardless of what behavior preceded it and for the remainder of the investigation we consider both isolated and escape-like turns together. Turning continued to modulate the likelihood of mechanosensory-evoked reversals even after animals had been stimulated multiple times and begun showing signs of habituation S1 Fig. And the probability of evoked reversals did not change appreciably in new experiments with modest changes of the inter-stimulus interval as shown in S2 Fig. And we show that light evoked reversals require the necessary optogenetic co-factor all-trans retinal, as expected, S3 Fig. In the remainder of the work, we present results from only new experiments designed to investigate how turning modulates mechonsensory-evoked reversals.

## Turns decrease the likelihood of interneuron-evoked reversals, except for those evoked by AVA

Mechanosensory signals from the anterior gentle-touch mechanosensory neurons AVM and ALM are thought to evoke a reversal response by traveling downstream through a network of interneurons that are associated with backward locomotion [15,16,21,30–33]. These include neurons AVD [16,30,34,35], AVA [36–38], AIZ [39], RIM [37,40], AIB [37], AVE [41] (Fig 2A, taken directly from nemanode.org [42]). Like the anterior mechanosensory neurons, interneurons AVA, AIZ, RIM, AIB, and AVE are known to induce reversals upon stimulation [37,39,41]. To better understand where this network interacts with turning, we sought to investigate whether these interneurons' ability to evoke reversals also depends on turning. We used a collection of transgenic strains with cell-specific or near-cell-specific promoters that drive expression of the optogenetic proteins Chrimson or ChR2 in each of these interneurons (Table 1). We then used our previously reported high-throughput closed-loop optogenetic delivery system [19] to stimulate the interneuron with 3 s whole-body illumination when the worm was either crawling forward or beginning to turn. In this way, we measured the animal's response to many thousands of optogenetic stimulation events.

As expected, optogenetic activation during forward locomotion of any of the interneurons AVE, AIZ, RIM, AIB, or AVA evoked reversals (Fig 2B) at a higher rate than the baseline probability of a spontaneous reversal (S4 Fig). Activating any of the interneurons we tested, except for AVA, showed a statistically significant decrease in the probability of evoking reversals when activated during turns, compared to during forward locomotion, Fig 2B. In other words, activation of these interneurons showed a turning-dependent response, similar to the mechanosensory neurons. By contrast, turning did not significantly modulate AVA's ability to evoke reversals and the worm often aborted its turn and reversed when AVA was activated during the turn (Fig 2B, S4 Video).

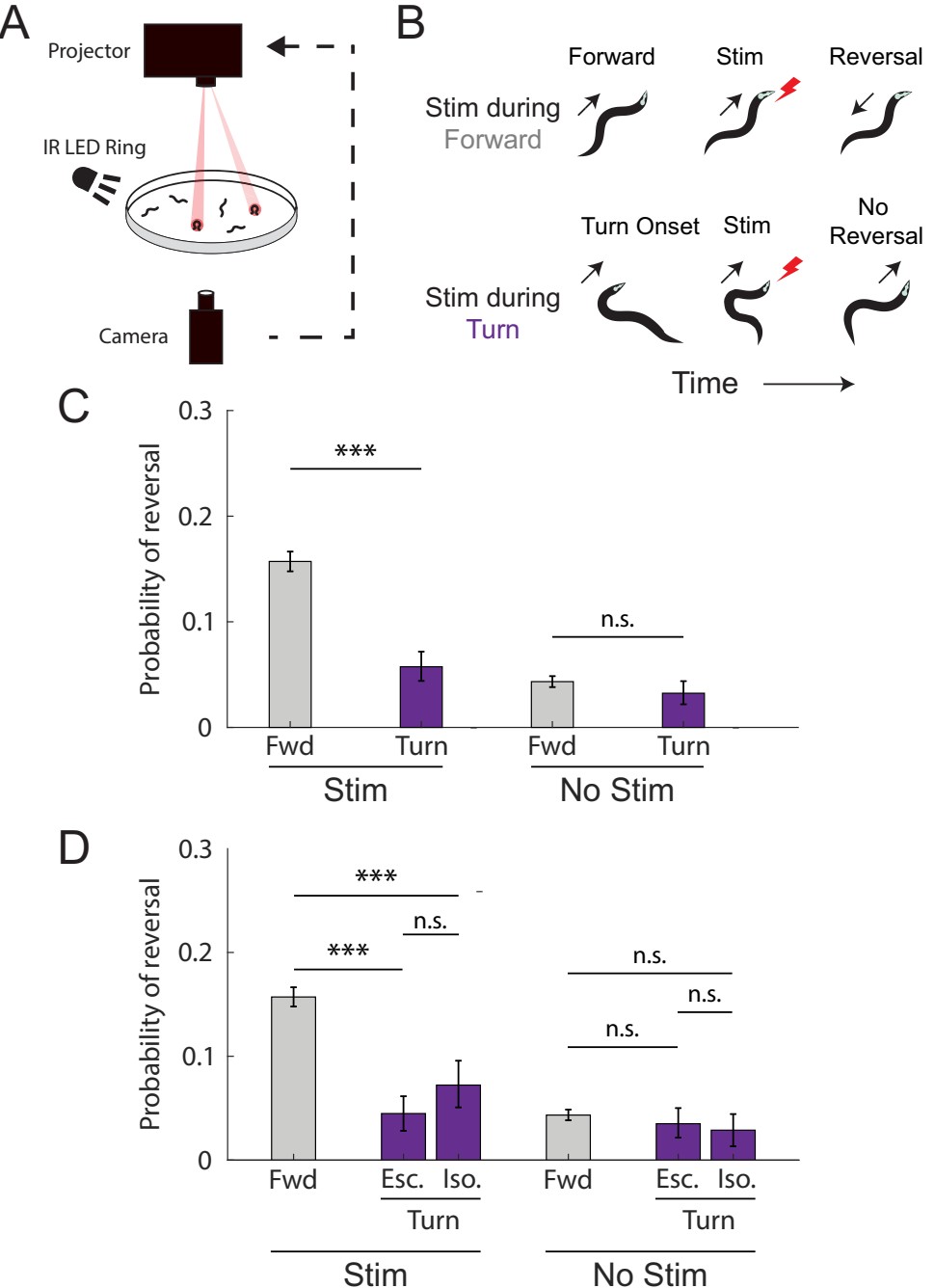

**Fig 1. Turns decrease the likelihood of mechanosensory-evoked reversals.** (A) Closed-loop optogenetic stimulation is delivered to animals as they crawl based on their current behavior. (B) Optogenetic stimulation is delivered to gentle-touch mechanosensory neurons in worms that are either moving forward (top row) or turning (bottom row). (C) The probability of a reversal is shown in response to stimulation during forward movement or turn. Responses are also shown for a low-light no-stimulation control. This figure only is a reanalysis of recordings from [19]. The number of stimulation events, from left to right: 6,002, 1,114, 5,996, and 1,050. (D) The probability of reversal in response to stimulation during turning is shown broken down further by turn subtype: escape-like turns "Esc" and isolated turns "Iso." $N$ = 6,002, 602, 512, 5,996, 599, and 451 stim events, from left to right. The number of plates for forward and turn context are 29 and 47, respectively. The 95% confidence intervals for population proportions are reported; *** indicates $p<0.001$, "n.s." indicates $p>0.05$ via two proportion Z-test. Exact $p$ values for all the statistical tests are listed in S1 Table. All data underlying this figure can be found at https://doi.org/10.25452/figshare.plus.23903202.

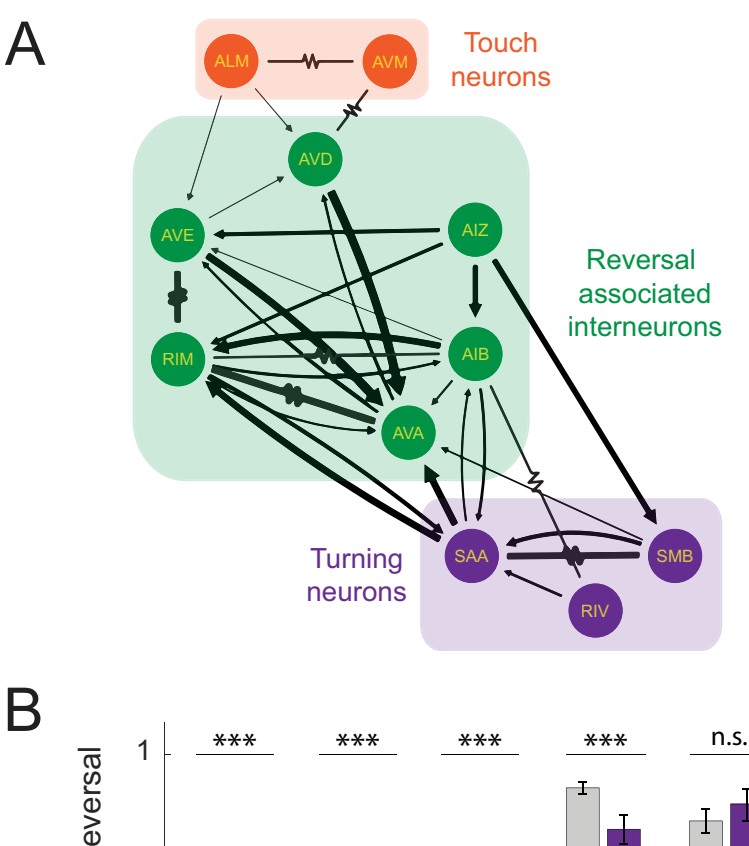

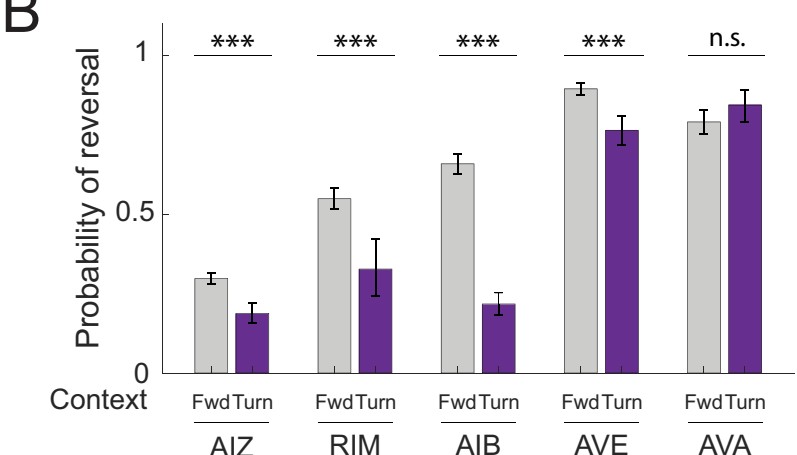

**Fig 2. Turns decrease the likelihood of interneuron evoked reversals, except for AVA.** (A) Anatomical connectivity showing chemical (arrows) and electrical (resistor symbol) synapses among the anterior mechanosensory neurons, downstream interneurons, and turning-associated neurons. (B) Probability of a reversal response is shown for 3 s optogenetic stimulation to the listed neurons either during forward movement or immediately after the onset of turning. Strains are listed in Table 1. Illumination was 80 $\mu$W/mm$^2$ red light to activate Chrimson in AVE or AVA, 300 $\mu$W/mm$^2$ blue light to activate ChR2 in RIM or AIB, and 340 $\mu$W/mm$^2$ to activate ChR2 in AIZ. Error bars indicate 95% confidence intervals for population proportions; *** indicates $p<0.001$, "n.s." indicates $p>0.05$ via two-proportion Z-test, and $p$ value for AVA stimulation group is 0.125. Exact $p$ values for all the statistical tests are listed in S1 Table. $N = 2,612, 601, 883, 107, 880, 511, 1,007, 342, 409,$ and 191 stimulus events, from left-to-right, measured across the following number of plates: 16, 27, 12, 19, 4, 24, 8, 16, 8, and 20. All data underlying this figure can be found at https://doi.org/10.25452/figshare.plus.23903202.

From these perturbations, we conclude that neurons AIZ, RIM, AIB, and AVE lie either at or upstream of the junction in which turning signals modulate the reversal response. AVA, in contrast, lies in the pathway downstream of the arrival of turning related signals. We therefore sought to investigate neural sources of this turning-related signal.

**Table 1. Strains used.**

| Strain name | Target neuron expression | additional expression | Genotype | Figure | Ref |
|---|---|---|---|---|---|
| AML67 | ALML, ALMR, AVM, PLML, PLMR, PVM | | wtfIs46[Pmec-4::Chrimson::SL2::mCherry::unc-54 40ng/ul] | Figs 1C, 1D, S1, S2, S3, S7B, and S8 | [18] |
| TQ3301 | AIZ | | xuIs198[Pser-2(2)::frt::ChR2::YFP,Podr-2(2b)::flp, Punc-122::YFP]; lite-1(xu7)X | Figs 2B, S4, and S8 | [39] |
| QW910 | RIM | | zfIs9[Ptdc-1::ChR2::GFP, lin-15+]; lite-1(ce314)X | Figs 2B, S4, and S8 | [40] |
| QW1097 | AIB | | zfIs112[Pnpr-9::ChR2::GFP, lin15+]; lite-1(ce314)X | Figs 2B, S4, and S8 | [40] |
| Not provided | AVE | | Popt-3::Chrimson | Figs 2B, S4, and S8 | [41] |
| AML17 | AVA | I1, I4, M4, and NSM [59] | wtfIs2[Prig-3::Chrimson::SL2::mCherry] | Figs 2B, S4, and S8 | This work |
| AML496 | RIV, SMB, SAA | | wtfIs465 [Plim-4::gtACR2::SL2::eGFP::unc-54 80ng/ul + Punc-122:: RFP 50ng/ul] | Figs 3, S5, S7C, and S8 | This work |
| AML499 | RIV, SMB, SAA; ALML, ALMR, AVM, PLML, PLMR, PVM | | wtfIs46[Pmec-4::Chrimson::SL2::mCherry::unc-54 40ng/ul]; wtfIs465 [Plim-4::gtACR2::SL2::eGFP::unc-54 80ng/ul + Punc-122::RFP 50ng/ul] | Figs 4, S6, S7A, and S8 | This work |
| N2 | - | | - | S8 Fig | |
| KG1180 | - | | lite-1(ce314) | S8 Fig | [61] |

We note that for any given perturbation shown in Fig 2B, we are interested in the change of probability of reversal between the forward and turning contexts. We do not concern ourselves with overall differences in reversal probability for perturbations of different neurons because that may arise from differences in gene expression or differences in the efficiencies of ChR2 compared to Chrimson. Stimulation of AVD was not tested because no suitable single-cell promoter was found.

## Turning-associated neurons RIV, SMB, and SAA regulate reversals

Turning in the worm occurs either when the animal is moving forward, is paused, or is transitioning from backward to forward locomotion, but not during sustained backward locomotion [43]. Neuron cell types RIV, SMB, and SAA are among those neurons associated with turning. RIV, SMB, and SAAD have increased calcium activity during turns [21,44], and ablation of RIV, SMB, or SAA show defects in turning or head bending amplitude [30,44]. Wang and colleagues observed that inhibiting RIV, SMB, and SAA when the animal is backing up prolongs the reversal [21]. They therefore proposed that activity from turning-related neurons may inhibit reversals. We expressed the blue light inhibitory opsin, gtACR2 [45,46], in these neurons and independently confirmed that inhibiting RIV, SMB, and SAA, increases reversal duration, Figs 3 and S6 Fig. We therefore sought to investigate whether these turning neurons also inhibit reversals during turns and whether they may explain why mechanosensory stimulation is less likely to evoke reversals during turning.

## Inhibiting RIV, SMB, and SAA abolishes the turning dependent modulation of mechanosensory processing

We reasoned that if the turning neurons RIV, SMB, and SAA inhibit reversals, then releasing this inhibition after a turn has begun should allow mechanosensory stimuli delivered during the turn to evoke reversals as effectively as if they were delivered during forward locomotion. We designed an experiment to simultaneously inhibit these turning neurons while stimulating the touch neurons immediately after the onset of a turn. We expressed a blue light inhibitory

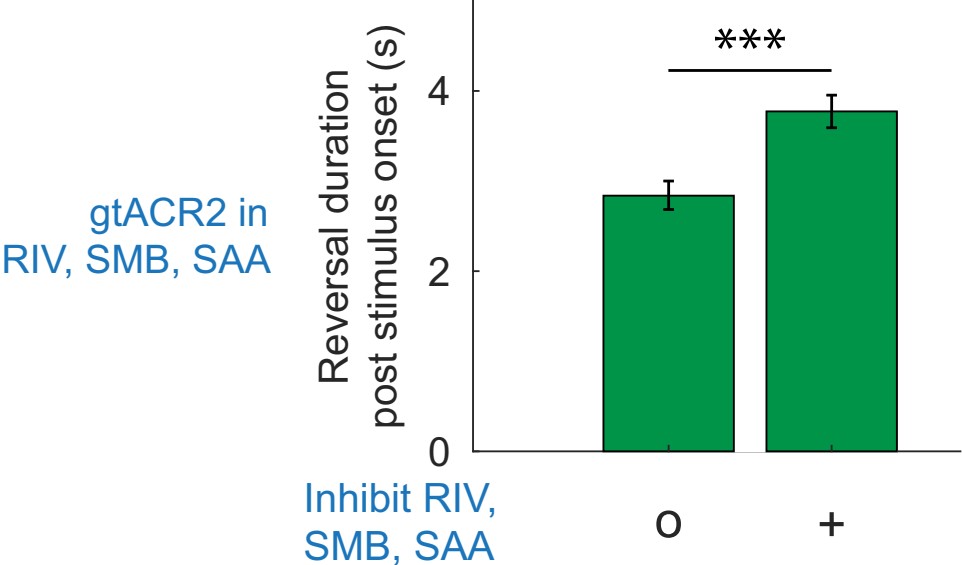

**Fig 3. RIV, SMB, and SAA neurons influence reversal duration.** Neurons RIV, SMB, and SAA were optogenetically inhibited when worms spontaneously reversed. The time spent going backwards is reported in a 10-s window coinciding with optogenetic inhibition upon reversal onset. Worms expressed the inhibitory opsin gtACR2 in neurons RIV, SMB, and SAA under the *lim-4* promoter. Illumination intensity of either 180 $\mu$W/mm$^2$ ("+") or 2 $\mu$W/mm$^2$ ("o" control) was delivered. Worms spent more time reversing when these neurons were inhibited than in the control. Error bars represent 95% confidence intervals; *p* value via two-proportion Z-test is 1.93$E$−09. *N* = 612 and 695 stimulus events for "o" and "+" conditions, respectively, across 14 plates. All data underlying this figure can be found at https://doi.org/10.25452/figshare.plus.23903202.

opsin, gtACR2, in the turning-associated neurons RIV, SMB, and SAA and a red light activating opsin Chrimson in the gentle-touch neurons. Inhibiting RIV, SMB, and SAA after the onset of a turn did not completely stop the animal and it still successfully exited the turn (see S5 Video). We reasoned that ongoing RIV, SMB, and SAA activity was not necessary for the completion of the turn once initiated and this therefore allowed us to inhibit these turning-associated neurons in a context in which the animal was still turning.

Activating the touch neurons by delivering red light immediately after the onset of a turn was less likely to evoke a reversal than when delivered during forward locomotion, Fig 4, as expected. But when we also inhibited the RIV, SMB, and SAA turning-associated neurons with blue light immediately after the turn began, the likelihood of evoking reversals via red light activation of the touch neurons was significantly higher and, crucially, not significantly different than for activation during forward locomotion (see S6 Video). In other words, inhibiting these turning-associated neurons after turn onset abolished the turning-dependence of the mechanosensory response. This is consistent with a model in which signals from RIV, SMB, and/or SAA disrupt mechanosensory processing during turning. By inhibiting those neurons after the onset of a turn, we prevent this disruption, presumably by inhibiting an inhibitory signal.

We performed additional experiments to rule out alternative explanations for why blue light illumination restored the likelihood of a mechanosensory-evoked reversal response (S7B Fig and S1 Text). For example, we find that blue light illumination when no inhibitory opsin is present is insufficient to restore mechanosensory-evoked reversal responses during turns, suggesting that the effect is not an artifact of the blue light alone (S7B Fig). Taken together, we conclude that inhibition of the turning neurons during turns disinhibits the mechanosensory-evoked reversal response.

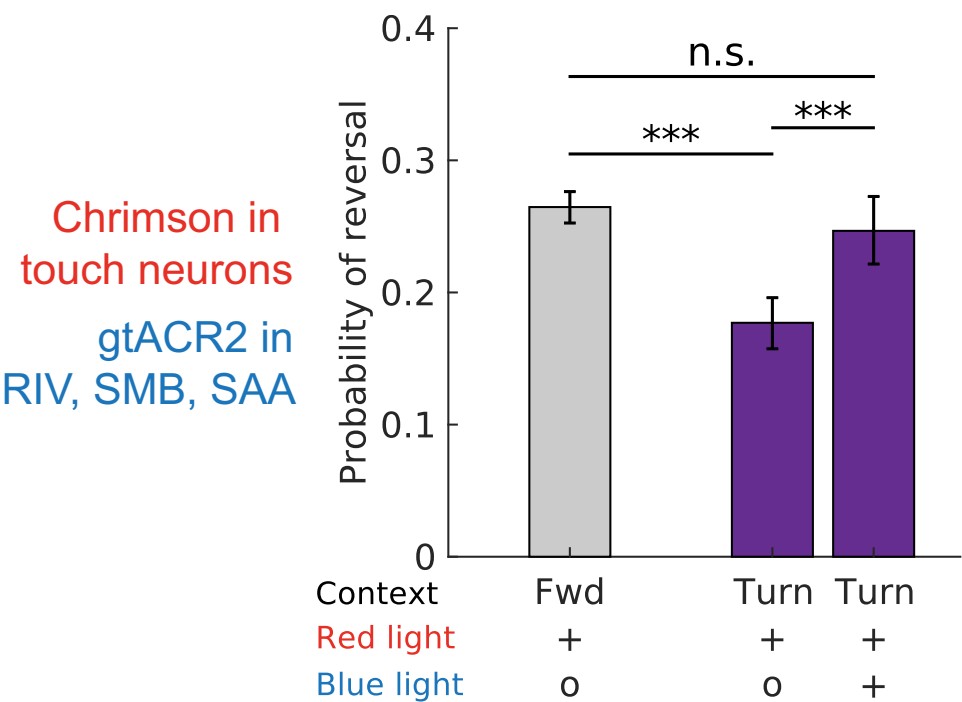

**Fig 4. Optogenetic inhibition of neurons RIV, SAA, and SMB during turns restore mechanosensory-evoked reversal response.** Probability of reversals when touch neurons are activated or when touch neurons are activated and RIV, SMB, and SAA are inhibited simultaneously, during either forward movement or turn onset. Touch neurons express Chrimson and are activated with red light. RIV, SMB, and SAA express gtACR2 and are inhibited with blue light. Strains are listed in Table 1. The 95% confidence intervals for population proportions are reported; *** indicates $p < 0.001$, "n.s." indicates $p > 0.05$ via two-proportion Z-test. Exact $p$ values for all the statistical tests are listed in S1 Table. $N$ = 5,381, 1,525, and 1,115 stimulation events from left to right. The number of plates from left to right bars are: $N$ = 8, 16, and 16. Additional controls are shown in S7 Fig. All data underlying this figure can be found at https://doi.org/10.25452/figshare.plus.23903202.

## Signals from turning neurons gate mechanosensory processing

Our measurements supports a model in which the turning neurons RIV, SMB, and/or SAA gate mechanosensory information and prevent it from propagating further downstream to evoke a reversal, Fig 5. In this model, mechanosensory signals from the gentle-touch mechanosensory neurons ALM and AVM propagate downstream in a feedforward manner to reversal-associated interneurons RIM, AIZ, AIB, and AVE. If the animal is moving forward, the mechanosensory signals continue to propagate to AVA and evoke reversals. But if the animal is turning, inhibitory signals originating from RIV/SMB/SAA suppress or disrupt mechanosensory-related signals within the interneurons and prevent downstream mechanosensory-related signals from propagating to AVA. This model is consistent with our measurements and leads us to conclude that turning-related inhibitory signals gates downstream mechanosensory processing.

## Discussion and conclusions

Here, we show that putative inhibitory signals from turning-associated neurons RIV/SMB/SAA modulate mechanosensory-evoked reversals downstream of the gentle-touch neurons and upstream of neuron AVA. But within those constraints, where exactly might those signals combine? Neuron wiring and gene expression data suggests that one location may be across the inhibitory synapses from SAA to AIB and RIM. SAA releases acetylcholine and makes

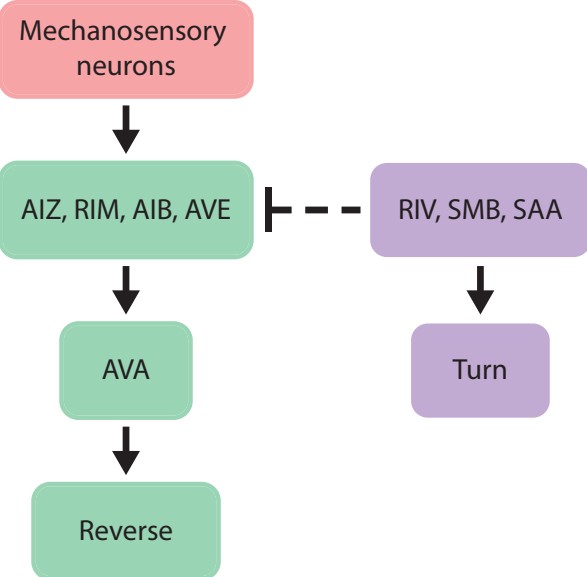

**Fig 5. Putative circuit mechanism.** In response to gentle-touch, mechanosensory neurons propagate signals downstream through the network and reach neuron AVA to evoke a reversal. But during turning, neurons RIV, SMB, and/or SAA send inhibitory signals that disrupt sensory-related signals before they reach AVA, thus gating the likelihood of a reversal.

chemical synapses onto AIB and RIM, which both express inhibitory acetylcholine receptors [42,47–50]: AIB expresses the inhibitory acetylcholine receptors *lgc-47*, and *acc-1*; while RIM expresses inhibitory (e.g., *lgc-47* [50], and *acc-1* [51]) and excitatory (e.g., *acr-3*) acetylcholine receptors. We note that AIB and RIM both synapse onto AVA, therefore, SAA-mediated inhibition of AIB and RIM may decrease overall excitation to AVA, broadly consistent with our cartoon model in Fig 5.

Wang and colleagues had previously predicted that turning circuitry may inhibit reversal circuitry [21]. Now in contemporaneous work from the same group, Huo and colleagues show that activation of SAA/RIV/SMB terminates reversals and inhibits RIM when RIM is already active [51] likely through an ACC-1 acetylcholine-gated chloride channel [51], but possibly also through LGC-47 [52].

Our findings are consistent with the mechanism proposed in [51] in which SAA blocks reversals by inhibiting RIM. More broadly our findings reinforce a longstanding hypothesis that different motor programs in the worm inhibit one another, as was previously proposed for forward and reverse locomotion [53].

In our model, AVA performs a role similar to that of a "decision neuron" with respect to reversals [54]. This is consistent with our previous observation that AVA's calcium activity more closely reflects the animal's decision to reverse and is less reflective of the strength of the stimulus (e.g., AVA's activity does not reflect how many touch neurons are activated) [36].

The simple model we describe in Fig 5 assumes feed-forward propagation of signals from ALM and AVM through the downstream network to AVA and omits recurrent connections among the neurons in between. Future investigations are needed to explore additional contributions from recurrence in the network, further complexities from wiring, such as potentially excitatory synaptic input from SAAV to AVA [49,55], and the role of AVD, for which we lacked a cell-specific promoter.

More broadly, we show that motor-related signals are directly influencing neural activity in areas that contain a mix of sensory and motor information. This is reminiscent of saccadic suppression in vision [56–58] and corollary discharge [27–29] in which motor-related activity modulates or impinges upon sensory representations. Our findings add to a growing body of evidence suggesting that behavior information is necessary for sensory processing. The brain's presumed need to access both types of information in the same place may explain why behavior-related neural activity patterns are seen across so many brain areas in mice, fly, and worms, including in nominally sensory areas [22–25].

Because turning events are infrequent, spontaneous and brief, they are rare compared to the time the animal spends moving forward or backwards. But obtaining sufficient statistical power to probe sensory processing during turns required hundreds of observations per condition. In total, we measured 97,268 behavior responses to stimulation, including 16,544 during turns. This investigation was therefore only made feasible by leveraging the recent high-throughput methods we presented in [19] that use computer vision and targeted illumination to track many worms in parallel and to automatically deliver stimuli triggered upon the animal's turns.

## Materials and methods

### Strains

Strains used in this work are listed in Table 1. Light-gated ion channels have been expressed in most strains to either excite or inhibit specific neurons. We expressed excitatory opsin Chrimson in the six gentle-touch neurons using the *mec-4* promoter. Promoters *ser-2*, *tdc-1*, *npr-9*, *opt-3*, and *rig-3* are used to express excitatory opsin in neurons AIZ, RIM, AIB, AVE, and AVA, respectively. Some strains have additional expression in other neurons, listed in Table 1. For example, the promoter *rig-3* is widely used to study AVA [36–38], as it is here, despite also having off-target expression in pharyngeal neurons I1, I4, M4, and NSM neurons [59]. To express gtACR2 in RIV, SMB, and SAA, we used the *lim-4* promoter (RRID: Addgene_195853), following the same strategy as in [21] and confirmed the expression pattern using fluorescence microscopy, S5 Fig. For that strain, we performed integration using a mini-SOG approach. We injected into CZ20310 worms, followed by a blue light treatment (450 nm, M450LP1, Thorlabs) for 30 min as described in [60]. Before conducting experiments, we out-crossed integrated worms with the wild-type N2 strains for at least six generations to generate AML496. AML496 worms were then crossed into AML67 worms to create AML499. Our transgenic strains include a mix of WT and *lite-1* mutant backgrounds. We measured no systematic difference in locomotion or to endogenous blue light response in these two backgrounds for the light levels and conditions used here S8 Fig.

### Nematode handling

All worm strains were maintained at 20°C, on regular NGM media plates seeded with *E. coli* (OP50) as food source. Experiments were performed on young adult animals. To obtain young adults, worms were bleached 3 days prior to the experiments. Bleached eggs were washed and centrifuged in M9 (0.8 rcf for 2 min) three times. Bleached eggs were suspended in M9 and stored in a shaker overnight. The following morning hatched L1 larvae were centrifuged and transferred to freshly seeded plates consisting of 1 ml of 0.5 mM all-trans-retinal mixed with OP50 and stored in the dark at 20°C until young adulthood.

For experiments, young adult worms were washed in M9 and transferred to an empty agarose plate for experiments. Excess M9 solution was absorbed with a kimwipe as described in [18,19].

## Behavior analysis

Computer vision-based behavior analysis was used to identify when the animal is moving forward, when it is undergoing a reversal, or when it is turning. The closed loop latency from detecting a turn to delivering an optogenetic stimulation is 167 ms [19]. Analysis was performed as reported previously using two different sets of algorithms, one for real time applications and the other retrospectively in post-processing [19]. All figures in this work reflect behavior classifications from the off-line retrospective analysis.

Briefly, animals are segmented and a centerline is detected. Additional logic is used to find centerlines even when the animal touches itself [18]. The animal's center of mass velocity is also computed. Behavior classification is first performed by classifying pose dynamics in a behavior map [18,62] and then refined by inspecting the animal's ellipse ratio and center of mass velocity to catch any omitted turns, or instances when the behavior mapper fails to classify. Compared to our previous recent work [19], we changed two parameters to be more conservative in classifying animals as turning or reversing. Specifically, to be classified as turning we now require that the binary image of the animal have an ellipse ratio of 3.1, compared to 3.6 previously. Similarly, to be classified as a reversal, the animal must now achieve a center of mass velocity of −0.11 mm/s, instead of −0.1 mm/s, during the 3 s optogenetic stimulus window. These changes were minor and were implemented to catch rare events that previously had escaped classification.

For experiments probing reversal duration, we report the time the animal spent going backwards in a 10 s window, coinciding with optogenetic inhibition; 10 s was chosen because it was a compromise between the 12 s used in [21] and the shorter stimuli that we typically use [19]. So for example, if after stimulus onset the animal continued moving backwards for 3 s, then paused for 1 s, and moved backwards for 2 s more, we report a "reversal duration" of 5 s.

## Optogenetic activation and inhibition

In this work, we deliver whole-body optogenetic illumination specifically when the animal is either moving forward, or turning, or reversing. We conduct different sets of experiments for each of these three conditions, using different sets of animals for each experiment. In all cases, we use a projector-based illumination system that tracks many individuals on a plate full of animals, segments them in real time, and addresses each animal individually to shine light on their whole body, as described previously [19]. All experiments are performed on plates containing approximately 30 to 40 animals.

To measure the animal's response to optogenetic activation or inhibition delivered during the onset of turns, our system waited until it detected that an animal was beginning to turn, and then delivered a stimulus automatically. In post-processing, we retrospectively evaluated whether the turn was valid at time of stimulus onset and only included those stimuli events that met our more stringent criteria, as described in [19].

To measure the animal's response to optogenetic perturbations during forward locomotion, we optogenetically illuminated all tracked animals on the plate every 30 s, in open loop. In post-processing, we then only considered those animals that were moving forward at the time of illumination. The worms in the open loop assays were stimulated every 30 s. However, in the closed loop experiments, the worms were stimulated when turns were detected. As a result, the worms received optogentic stimulus less frequently, shown in S2B Fig. There was no statistically significant difference in the probability of evoked reversal for stimuli delivered during forward locomotion in these two conditions S2A Fig.

To measure the animal's response to optogenetic inhibition during reversals, our system waited until it detected that an animal had been reversing for 1 s, and then delivered the

illumination. As before, we retrospectively confirmed that the animal was reversing before including it for further analysis.

Illumination color, intensity, and duration are listed in Table 2.

**Table 2. List of optogenetic measurements performed during behavior.**

| Target neuron(s) | Perturbation | Target behavior | Stim triggered on | Stim duration (s) | ISI (s) | Illumination intensity (µW/mm²) | Illumination color | Strain | ATR plates | # Plates | Total stim events | Figures | Ref. |
|---|---|---|---|---|---|---|---|---|---|---|---|---|---|
| ALML, ALMR, AVM, PLML, PLMR, PVM | Excite Chrimson | Forward | - | 3 | 30 | 0.5, 80 | Red | AML67 | + | 29 | 11,998 | Figs 1C, 1D, S1, S2B, and S3 | [19] |
| | | Turn | Turns | | >30 | | | | + | 47 | 2,164 | | |
| AIZ | Excite ChR2 | Forward | - | 3 | 30 | 2, 340 | Blue | TQ3301 | + | 16 | 5,258 | Figs 2B, S4, and S8A | This work |
| | | Turn | Turns | | >30 | | | | + | 27 | 1,184 | | |
| RIM | Excite ChR2 | Forward | - | 3 | 30 | 2, 300 | Blue | QW910 | + | 12 | 1,766 | Figs 2B, S4, and S8A | This work |
| | | Turn | Turns | | >30 | | | | + | 19 | 238 | | |
| AIB | Excite ChR2 | Forward | - | 3 | 30 | 2, 300 | Blue | QW1097 | + | 4 | 1,747 | Figs 2B, S4, and S8A | This work |
| | | Turn | Turns | | >30 | | | | + | 24 | 1,038 | | |
| AVE | Excite Chrimson | Forward | - | 3 | 30 | 0.5, 80 | Red | AVE | + | 8 | 2,413 | Figs 2B and S4 | This work |
| | | Turn | Turns | | >30 | | | | + | 16 | 832 | | |
| AVA | Excite Chrimson | Forward | - | 3 | 30 | 0.5, 80 | Red | AML17 | + | 8 | 1,035 | Figs 2B and S4 | This work |
| | | Turn | Turns | | >30 | | | | + | 20 | 411 | | |
| RIV, SMB, SAA | Inhibit gtACR2 | Reversal | Reversals | 10 | >30 | 2, 180 | Blue | AML496 | + | 14 | 1,307 | Fig 3 | This work |
| ALML, ALMR, AVM, PLML, PLMR, PVM, RIV, SMB, SAA | Inhibit gtACR2 | Reversal | Reversals | 10 | >30 | 2, 180 | Blue | AML499 | + | 12 | 2,532 | S6 Fig | This work |
| ALML, ALMR, AVM, PLML, PLMR, PVM, RIV, SMB, SAA | Excite Chrimson | Forward | - | 3 | 30 | 60 | Red | AML499 | + | 8 | 5,381 | Figs 4 and S7A | This work |
| | | Turn | Turns | | >30 | | | | + | 16 | 1,525 | | |
| | Excite Chrimson and Inhibit gtACR2 | Turn | Turns | | >30 | Red = 60, Blue = 180 | Red + Blue | | + | 16 | 1,115 | | |
| | Inhibit gtACR2 | Turn | Turns | | >30 | 180 | Blue | | + | 15 | 954 | | |
| | Control | Turn | Turns | | >30 | Red = 0.5, Blue = 2 | Red + Blue | | + | 8 | 1,961 | | |
| ALML, ALMR, AVM, PLML, PLMR, PVM | Excite Chrimson | Forward | - | 3 | 30 | 60 | Red | AML67 | + | 6 | 3,722 | S7B Fig | This work |
| | | Turn | Turns | | >30 | | | | + | 12 | 903 | | |
| | | Turn | Turns | | >30 | Red = 60, Blue = 180 | Red + Blue | | + | 15 | 794 | | |
| | | Turn | Turns | | >30 | 180 | Blue | | + | 16 | 579 | | |
| | Control | Turn | Turns | | >30 | Red = 0.5, Blue = 2 | Red + Blue | | + | 15 | 772 | | |
| RIV, SMB, SAA | Inhibit gtACR2 | Turn | Turns | 3 | >30 | 2, 180 | Blue | AML496 | + | 16 | 2,074 | S7C Fig | This work |

*(Continued)*

**Table 2.** (Continued)

| Target neuron(s) | Perturbation | Target behavior | Stim triggered on | Stim duration (s) | ISI (s) | Illumination intensity (µW/mm²) | Illumination color | Strain | ATR plates | # Plates | Total stim events | Figures | Ref. |
|---|---|---|---|---|---|---|---|---|---|---|---|---|---|
| ALML, ALMR, AVM, PLML, PLMR, PVM | Control to test Endogonous Blue Light Response | Forward | - | 3 | 30 | 300 | Blue | AML67 | - | 4 | 6,564 | S8A Fig | This work |
| AIZ | | Forward | - | 3 | 30 | 300 | Blue | TQ3301 | - | 4 | 3,213 | | This work |
| RIM | | Forward | - | 3 | 30 | 300 | Blue | QW910 | - | 4 | 3,365 | | This work |
| AIB | | Forward | - | 3 | 30 | 300 | Blue | QW1097 | - | 4 | 3,867 | | This work |
| AVE | | Forward | - | 3 | 30 | 300 | Blue | AVE | - | 4 | 7,006 | | This work |
| AVA | | Forward | - | 3 | 30 | 300 | Blue | AML17 | - | 4 | 993 | | This work |
| RIV, SMB, SAA | | Forward | - | 3 | 30 | 300 | Blue | AML496 | - | 4 | 4,516 | | This work |
| ALML, ALMR, AVM, PLML, PLMR, PVM, RIV, SMB, SAA | | Forward | - | 3 | 30 | 300 | Blue | AML499 | - | 4 | 3,324 | | This work |
| - | | Forward | - | 3 | 30 | 300 | Blue | N2 | - | 4 | 646 | | This work |
| - | | Forward | - | 3 | 30 | 300 | Blue | KG1180 | - | 4 | 6,470 | | This work |
| ALML, ALMR, AVM, PLML, PLMR, PVM | Excite Chrimson | Forward | - | 3 | 30 | 0.5, 80 | Red | AML67 | + | 4 | 1,631 | S2A Fig | This work |
| ALML, ALMR, AVM, PLML, PLMR, PVM | Excite Chrimson | Forward | - | 3 | 59 | 0.5, 80 | Red | AML67 | + | 4 | 1,094 | S2A Fig | This work |
| ALML, ALMR, AVM, PLML, PLMR, PVM | Excite Chrimson | Forward | - | 3 | 30 | 80 | Red | AML67 | - | 4 | 876 | S3 Fig | This work |
| | | | | | | | | | | 452 | 97,268 | | |

## Statistical analysis

In our analysis, stimulus events are the fundamental unit. Throughout the manuscript, we report the proportion of all stimulus events that result in a reversal, the total number of stimulus events, and the corresponding 95% confidence interval, calculated analytically. To reject

the null hypothesis that two empirically observed proportions are the same, we use a two-proportion Z-test and report a *p* value [63].

## Supporting information

**S1 Fig. Probability of reversing in response to stimuli delivered during turns is consistently lower than in response stimuli delivered during forward locomotion throughout the duration of the 30-min assay.** Probability of evoked reversal in response to optogenetic stimulation to gentle-touch mechanosensory neurons (*Pmec-4::Chrimson*) is calculated for three different portions of the 30-min experiment. Habituation is visible, but the relative difference in reversal probability persists. Error bars show 95% confidence intervals of the population proportions; *** indicates $p < 0.001$ via two-proportion Z-test. Exact *p* values for all the statistical tests are listed in S1 Table. *N* = 2,006, 420, 2,077, 403, 1,919, and 291 stimulation events from left to right. The number of assay plates for forward and turn context are *N* = 29 and 47, respectively. This figure is a reanalysis of measurements presented in [19]. All data underlying this figure can be found at https://doi.org/10.25452/figshare.plus.23903202.
(PDF)

**S2 Fig. Probability of reversal is similar for two different inter-stimulus intervals.** (A) Animals expressing Chrimson in their gentle-touch mechanosensory neurons were optogenetically stimulated in open loop every 30 s or 59 s. Only responses to stimuli delivered during forward locomotion are included. These are new experiments not previously reported. *N* = 1,631 and 1,094 stim events for 30 s and 59 s inter-stimulus interval assays. We used four plates for both 30 s and 59 s inter-stimulus interval assays. Error bars show 95% confidence intervals of the population proportions, and *p* value via two-proportion Z-test is 0.196. (B) 59 s (vertical red bar) is the mean inter stimulus interval (ISI) experienced by worms in the closed-loop turn-triggered stimulus experiments previously presented in [19]. The ISI is not constant because it depends on when the worm turns. The distribution of the ISI experienced by worms during those experiments in Fig 1 is shown in blue. All data underlying this figure can be found at https://doi.org/10.25452/figshare.plus.23903202.
(PDF)

**S3 Fig. Red light evoked reversal responses are all-trans retinal dependent, as expected.** Animals that express chrimson in the touch receptor neurons were grown in the presence or absence of the necessary co-factor all-trans retinal (ATR) and exposed to 80 $\mu$W/mm$^2$ intensity red light. The 95% confidence intervals for population proportions are reported. Two sample Z-test was used to calculate significance; *** indicates $p < 0.001$. The exact *p* value is listed in S1 Table. The number of stimulus events for each condition (from left bar to right bar) are: 6,002 and 876. The number of assay plates for ATR + and ATR − conditions are 29, 4. Note that the ATR + condition was previously reported in [19] and also appears in Fig 1C and 1D. The ATR − condition was recorded contemporaneously, but is presented here for the first time. All data underlying this figure can be found at https://doi.org/10.25452/figshare.plus.23903202.
(PDF)

**S4 Fig. Baseline reversal probabilities measured via low-light (control) illumination.** These are control experiments corresponding to the experiments presented in Fig 2B. Baseline reversal probabilities for each strain in each condition are measured by shining a low-intensity control stimulus. Three seconds of only 0.5 $\mu$W/mm$^2$ of red light illumination (neuron AVE and AVA) or 2 $\mu$W/mm$^2$ of blue light illumination (neuron AIZ, RIM, and AIB). The 95% confidence intervals for population proportions are reported. Two proportion Z-test was used to calculate significance; *p* value for AIZ, RIM, AIB, AVE, and AVA stimulation group is 0.596,

0.936, 0.045, 0.565, 0.262, respectively. The number of stimulus events for each condition (from left-most bar to right-most bar) are: 2,646, 583, 883, 131, 867, 527, 1,406, 490, 626, and 220. The number of assay plates for forward and turn context for neurons from left to right are 16, 27, 12, 19, 4, 24, 8, 16, 8, and 20. All data underlying this figure can be found at https://doi.org/10.25452/figshare.plus.23903202.
(PDF)

**S5 Fig. Expression pattern of *lim-4* promoter.** Fluorescence/Bright field, merged image of AML496 worms showing the expression of eGFP driven by *lim-4* promoter using (*Plim-4*::*gtACR2*::*SL2*::*eGFP*) expression vector. eGFP can be seen in the neurons RIV, SMB, and SAA.
(PDF)

**S6 Fig. Inhibition of RIV, SMB, and SAA prolong reversals, in a second transgenic background.** Same experiment as in Fig 3, but in a transgenic background that also expresses Chrimson in the mechanosensory neurons. Results are consistent with Fig 3. Worm spent more time reversing when the RIV, SMB, and SAA neurons were inhibited compared to when a control stimulus intensity was used. Error bars represent 95% confidence intervals; *** indicates $p < 0.001$ via two-proportion Z-test. The exact $p$ value is listed in S1 Table. The number of stimulus events for mock and experimental conditions are 1,168 and 1,364, respectively. The number of assays was $N = 12$. All data underlying this figure can be found at https://doi.org/10.25452/figshare.plus.23903202.
(PDF)

**S7 Fig. Additional control experiments show that blue light alone cannot restore mechanosensory-evoked reversal response.** (A) Probability of reversals when either touch neurons are activated, or RIV, SMB, and SAA are inhibited, or both simultaneously; during either forward movement or turn onset. First three bars are same as in Fig 4. Touch neurons express Chrimson and are activated with red light. RIV, SMB, and SAA expressing gtACR2 are inhibited with blue light. Strains are listed in Table 1. The 95% confidence intervals for population proportions are reported. $N = 5,381, 1,525, 1,115, 1,961$, and 954 stim events, from left to right. The number of assays from left to right bars are: $N = 8, 16, 16, 8$, and 15. (B) Same experiments were repeated in a strain that expressed Chrimson in the gentle-touch mechanosensory neurons, but no inhibitory opsins. $N = 3,722, 903, 794, 772$, and 579 stim events. The number of assays from left to right bars are: $N = 6, 12, 15, 15$, and 16. (C) Same experiments are shown for animals that only express inhibitory opsin gtACR2 in RIV, SMB, and SAA, but no Chrimson. $N = 1,041$ and 1,033 stim events. The number of assay is: $N = 16$; *** indicates $p < 0.001$, "n.s." indicates $p > 0.05$ via two-proportion Z-test. Exact $p$ values for all the statistical tests are listed in S1 Table. All data underlying this figure can be found at https://doi.org/10.25452/figshare.plus.23903202.
(PDF)

**S8 Fig. Endogenous blue light sensitivity and baseline locomotion activity of strains used.** (A) To characterize endogenous sensitivity to blue light, blue light-evoked reversal probability is measured for different strains with and without the all-trans retinal (ATR) co-factor needed for optogenetic proteins. The 300 $\mu$W/mm$^2$ blue light intensity used here, is less than that reported to evoke the animal's endogenous blue light response [64]. Only those strains that express ChR2 are measured on retinal (ATR+, left, $N = 2,612, 883, 880$ from left to right, same as Fig 2B), while all strains, including the Chrimson strains, are measured in the off-retinal condition (ATR-, right, $N = 6,564, 3,213, 3,365, 3,867, 7,006, 993, 4,516, 3,324, 646$, and 6,470 from left to right). Error bars show 95% confidence intervals for population proportions. We include a *lite-1* mutant and wild-type N2 for comparison because our transgenic strains

include a mix of both wild-type and *lite-1* backgrounds. (B) Average speed of each strains used in this work are shown $N$ = 1,654, 564, 1,065, 654, 983, 1,099, 1,251, 837, 1,706, and 1,952 from left to right. All data underlying this figure can be found at https://doi.org/10.25452/figshare.plus.23903202.
(PDF)

**S1 Video. Example showing behavior of a population of animals during an experiment from [19].** Middle 24 s of a 30-min recording is shown. Optogenetic stimulation is delivered in closed loop when turning of an individual animal is detected. Each yellow numbered "x" represents a tracked animal, with its track shown in yellow. Inset at top left shows detailed movements of worm number 213, denoted by a green square. The head of the worm is represented by a green dot. A centerline is drawn through the worm's body and is shown in green. The dynamic circular pattern of green and white spots in the center of the video is a visual timestamp system projected onto the plate that is used for synchronizing the timing of video analysis, as described in [19].
(MP4)

**S2 Video. Example of a worm reversing in response to optogenetic stimulation of its gentle-touch mechanosensory neurons delivered during forward locomotion.** Recording is from [19]. Animals express Chrimson in gentle-touch mechanosensory neurons (strain name: AML67). Stimulus was delivered in open loop. Green dot denotes the animal's head. Green line denotes its centerline. Yellow line shows the trajectory of a point midway along the animal's centerline over the past 10 s. Red indicates area illuminated by red light.
(MP4)

**S3 Video. Example of a worm receiving optogenetic stimulation of its gentle-touch mechanosensory neurons during the onset of a turn.** Recording is from [19]. Animals express Chrimson in gentle-touch mechanosensory neurons (strain name: AML67). This worm does not reverse in response to stimulation. Stimuli was triggered in closed-loop by the animal's turn. Green dot denotes the animal's head. Green line denotes its centerline. Yellow line shows the trajectory of a point midway along the animal's centerline over the past 10 s. Red indicates area illuminated by red light.
(MP4)

**S4 Video. Example of a worm aborting a turn and reversing when neuron AVA was activated following the onset of the turn.** Animals express Chrimson in neuron AVA (strain name: AML17). Stimulation was delivered upon the onset of a turn in closed loop. Green dot denotes the animal's head. Green line denotes its centerline. Yellow line shows the trajectory of a point midway along the animal's centerline over the past 10 s. Red indicates area illuminated by red light.
(MP4)

**S5 Video. Example of a worm completing a turn during inhibition of neurons RIV, SMB, and SAA.** Animals express the inhibitory opsin gtACR2 in these neurons (strain name: AML496). Stimulation was delivered upon the onset of a turn in closed loop. Green dot denotes the animal's head. Green line denotes its centerline. Yellow line shows the trajectory of a point midway along the animal's centerline over the past 10 s. Blue indicates area illuminated by blue light.
(MP4)

**S6 Video. Example of a worm aborting a turn and reversing when neurons RIV, SMB, and SAA are inhibited and the gentle-touch mechanosenseory neurons are activated (strain**

**name: AML499).** Animals express the inhibitory opsin gtACR2 in RIV, SMB, and SAA and the excitatory opsin Chrimson in the gentle-touch mechanosensory neurons. Blue and red light illumination was delivered simultaneously upon the onset of a turn in closed loop. Green dot denotes the animal's head. Green line denotes its centerline. Yellow line shows the trajectory of a point midway along the animal's centerline over the past 10 s. Purple indicates area illuminated by red and blue light.
(MP4)

**S1 Table. Multi-sheet excel spreadsheet containing *p* values corresponding to all comparisons reported in all figures in manuscript.** Each figure corresponds to a different sheet; *p* values are calculated via two-proportion Z-test.
(XLSX)

**S1 Text. Supplementary text detailing the additional control experiments to show that blue light alone cannot restore the mechanosensory evoked reversals.**
(PDF)

## Acknowledgments

We thank Zhaoyu Li (Queensland Brain Institute), Shawn Xu (University of Michigan), and Mark Alkema (University of Massachusetts Worcester) for strains. We thank Matthew Creamer for helpful discussions. This work used computing resources from the Princeton Institute for Computational Science and Engineering. Strains from this work are being distributed by the CGC, which is funded by the NIH Office of Research Infrastructure Programs (P40 OD010440).

The content is solely the responsibility of the authors and does not represent the official views of any funding agency.

## Author Contributions

**Conceptualization:** Sandeep Kumar, Andrew M. Leifer.

**Formal analysis:** Sandeep Kumar, Andrew M. Leifer.

**Investigation:** Sandeep Kumar, Andrew Tran, Mochi Liu.

**Methodology:** Sandeep Kumar, Anuj K. Sharma.

**Project administration:** Andrew M. Leifer.

**Resources:** Anuj K. Sharma, Mochi Liu.

**Software:** Sandeep Kumar.

**Supervision:** Andrew M. Leifer.

**Visualization:** Sandeep Kumar.

**Writing – original draft:** Sandeep Kumar, Andrew M. Leifer.

**Writing – review & editing:** Sandeep Kumar, Andrew M. Leifer.

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
