## [Editor Report · Decision Letter 0]

28 Apr 2023

Dear Dr Leifer, 

Thank you for submitting your manuscript entitled "Inhibitory feedback from the motor circuit gates mechanosensory processing in C. elegans" for consideration as a Update Article by PLOS Biology.

Your manuscript has now been evaluated by the PLOS Biology editorial staff as well as by an academic editor with relevant expertise and I am writing to let you know that we would like to send your submission out for external peer review.

Once your full submission is complete, your paper will undergo a series of checks in preparation for peer review. After your manuscript has passed the checks it will be sent out for review. To provide the metadata for your submission, please Login to Editorial Manager (https://www.editorialmanager.com/pbiology) within two working days, i.e. by May 02 2023 11:59PM.

Kind regards,

Lucas

Lucas Smith, Ph.D.

Associate Editor

PLOS Biology

lsmith@plos.org

---

## [Decision Letter · Decision Letter 1]

23 Jun 2023

Dear Dr Leifer,

Thank you for your patience while your manuscript "Inhibitory feedback from the motor circuit gates mechanosensory processing in C. elegans" went through peer-review at PLOS Biology. Your manuscript has now been evaluated by the PLOS Biology editors, an Academic Editor with relevant expertise, and by several independent reviewers.

The reviews are appended at the end of this email, and you will see that all 4 reviewers are enthusiastic about the study, noting that it offers a useful contribution to the field and that it is generally well done. The reviewers have provided a number of suggestions to strengthen the message and framing of the study and we think their comments should be carefully addressed. Some of the reviewer comments invite additional experimental data - and we encourage you to add this data if you have it, or if you feel it will strengthen the study. However, after discussion with the Academic Editor, we would not strictly require additional experiments for publication. 

We are therefore pleased to offer you the opportunity to address the comments from the reviewers in a revision that we anticipate should not take you very long. We will then assess your revised manuscript and your response to the reviewers' comments with our Academic Editor aiming to avoid further rounds of peer-review, although might need to consult with the reviewers, depending on the nature of the revisions.

We expect to receive your revised manuscript within 1 month - but if you plan on conducting additional experiments and need more time, please let us know as we would be happy to extend the deadline for your revision. Please email us (plosbiology@plos.org) if you have any questions or concerns, or would like to request an extension. 

**IMPORTANT - As you address the reviewer requests, please also attend to the following editorial requests: 

1) TITLE: We encourage you to spell out the genus name in the title. ""Inhibitory feedback from the motor circuit gates mechanosensory processing in Caenorhabditis elegans"

2) DATA: Thank you for providing the raw data related to your manuscript on figshare. Can you please reference this dataset in each relevant figure legend? For example, you can add the sentence "the data underlying this figure can be found at https://doi.org/10.6084/m9.figshare.21699668"

** SUBMITTING YOUR REVISION**

*Resubmission Checklist*

*Published Peer Review*

*Blot and Gel Data Policy*

Sincerely,

Luke

Lucas Smith, Ph.D.

Senior Editor

PLOS Biology

lsmith@plos.org

REVIEWS:

Reviewer #1: In this study, the authors dissect the neural circuitry by which one C. elegans motor program (turning) influences sensorimotor transformations in another motor circuit (touch avoidance circuit). C. elegans has become a premier model for examining such problems, due to its well defined circuitry and the precision of the genetic/cellular perturbations that can be conducted. Building off of an elegant previous study, the authors here examine how defined turning neurons influence touch-elicited reversals. They first show that touch-induced reversals (elicited by opto stim of the touch sensory neurons) is inhibited both during reversal-associated turns and isolated turns (i.e. turns occurring outside of the context of the reversal behavioral sequence). Next, they perform the same experiment, but elicit the reversal at different nodes in the touch avoidance circuit. They find that the effect is maintained in some interneurons the lie downstream of the touch neurons, but not everywhere. For example, stimulation of AVA command neurons elicits reversals regarding of turning status. This maps plausible locations where the turning circuit may act on the reversal circuit. They then optogenetically inhibit the turning neurons RIV/SAA/SMB and first show that this causes animals to have longer reversals, as expected. Next they should that this inhibition is sufficient to reverse the effects of turning on touch-elicited reversals. They propose an intriguing model where these motor programs inhibit one another, such that sensorimotor transformations are gated by ongoing motor state.

This is an exciting study that makes elegant use of excellent technologies uniquely available in the authors' lab. The study builds off of previous ones in the field, but adds significant value, arriving at a clear model. My suggestions (mostly non-essential) are below:

1. I think it would be nice to cite the corollary discharge papers related to RIM (Ji, Riedl, etc) in the Introduction, as well as the Discussion (where they are currently cited). This will provide readers with proper background info in case they are not familiar with the field.

2. I think that a direct -/+ ATR comparison for the TRN::Chrimson really belongs in Fig. 1, rather than being buried in supplement (would be nice to show -/+ side by side also). This should also be shown for the strain/effect in Fig. 3 since that involves exaggerated reversals (which in principle could be blue light-stimulated).

3. The specificity of the lim-4 promoter used for SAA/SMB/RIV expression is not discussed. I think it would be important to show an image of the neurons that it labels and list out the neurons in the main text. In case there are any additional neurons, it would be useful for this to be recorded in the paper.

4. The induction of reversals by TRN::Chrimson is fairly weak compared to the interneuron stimulations. Can the authors comment on this? Have they attempted experiments at higher light intensities and are the effects of turning on stimulated reversals the same? (non-essential)

5. Does inhibition of RIV/SMB/SAA inhibit the induction of spontaneous reversals (in addition to extending reversal duration, as is shown in Fig. 3)?

6. In the dual-opto experiments, does the red light inhibit turning in a strain that only has lim-4::GtACR2? In other words, does the red light cross react with the blue opsin? Providing this type of information is useful for the field since other groups will likely become interested in dual-color optogenetics.

Reviewer #2: I am reviewing the paper ``Inhibitory feedback from the motor circuit gates mechanosensory processing in C.

elegans'' by Kumar et al. 

In this manuscript the authors investigate how turning during C. elegans locomotion influences the touch response on the interneuron level. They observed that optogenetic stimulation of TRN-specific channelrhodopsins did not elicit a robust escape response when the animal was in the process of performing turn behavior. The authors identified that the turning-associated motor neurons (SMB, RIV, SAA) suppress processing of the touch response on the interneuron level. These observation and deduced wiring diagram add significantly to the understanding of the systems-level regulation of a long-known sensory process. The research provides insight into more general mechanism of how motor-related signals can tune the neural dynamics, in the presence of mixture of contradictory sensory and motor signals. Thanks to automatizing the experimental pipeline, authors supported their hypotheses with the great amount of experimental data. The paper will greatly contribute to the field. I have several questions/comments.

I would suggest some improvements that should facilitate the understanding and its relation to the differentiate between sensory processing and sensitivity. 

If I am not mistaken, the process under study is related to corollary discharge, in which information from the motorcircuit or downstream interneurons adjust sensory processing, eg. prepare the nervous system for an incoming change that needs to be suppressed. I miss a deeper introduction to these concepts and earlier work/reviews on this topic and the authors should include timely citations (PMID: 35690069, 33880993). The authors cite some of these, including Crapse et al, but the discussion of the TRNs in this review is quite confusing and probably different from what the authors study here, thus they should explain what specifically they cite this review for. For a more general introduction to the body brain feedback I suggest PMID 35687973.

Is there evidence in general from the authors or in the literature that turns or body bends in general affect touch response?

The observation that turns modulate TRN processing asks the question if turns towards the ventral side and turns towards the dorsal side are similarly affected. AVM and PVM are located on the ventral side and are thus asymmetrically affected by body bends. Does exaggerated bending during turns directly modulate the sensitivity/excitability of the TRNs? To differentiate between passive body bending that affects the sensitivity vs inhibitory feedback from the interneurons, the authors could perform experiments in animals with exaggerated body curvatures.

From reading the manuscript I got the impression that the optogenetic stimulus was specifically patterned to activate anterior TRNs. but now I am not sure anymore. Posterior neurons are not discussed, but if the whole animal is illuminated how does optogenetic stimulation of the posterior touch neurons affect their results? Would PLM processing also be affected by turns, or would they still trigger escape and accelerate animal locomotion?

The turning associated motorneurons SMB, RIV and SAA were inhibited to activate the reversal response during turns. The rational was that they show increased Ca activity during turns in a freely behaving animal. Why have all three neurons been targeted and what is the response for each of them? Are they redundantly acting or is there a dominant player? 

The figures are a bit weak in general. The bar charts are not very informative with respect to sample distribution, and without raster plot I do not get a good idea about the habituation to the stimuli (In Figure S1, a reanalysis is provided though). The legend says N number of stimulations from x plates, but not how many animals and stimuli per animals have been scored. What is the central tendency? 95% CI on the mean, median? Bootstrapping suggest it is the median, but I needed to guess. Exact p-values are not given, neither the degrees of freedom for their calculation (based on animals, plates, stimulations?). As the authors use a z-test as opposed to a t-test, they use the number of stimulations for their statistical comparisons. One would wonder how the stimulus-response actually varied between individual plates and animals.

The authors should also expand their discussion on why this escape response needs to be suppressed. 

The authors state that: ``Suppressing mechanosensory-evoked reversals during turns may be part of a predator avoidance strategy. Turns are an important part of the C. elegans escape response, and by preventing turns from being interrupted prematurely, the animal may be ensuring that the escape response continues to completion.''. but this argument seems not very intruiging as the animal is might encounter a new threat which justifies a full sensory response. What I am trying to say is the the successful completion of the escape response into a new threat does not help the animal to survive. Is it expected that TRNs activate during omega turns? Is there a precedence for this? If so, this would make a strong physiological link between the motor command circuit and the suppression of a sensory stimulus. I don't think there is evidence in the literature that TRNs activate during omega turn, but if the authors had this data, it would make a strong point.

In the last decade, there has been some work supporting the hierarchical organisation of neural network in C. elegans, (e.g. PMID: 31786012), where forward-backward locomotion switch is at the highest organisational level (with AIB neuron as a representative), and with head bends/turns at lower organisational level. Can authors comment, even hypothetically, how their model relates to that? Is it contradictory to hierarchical organisation model?

The authors should discuss the limitations of their optogenetic approach, e.g. lack of physiological depolarization, no locally confined signals, e.g. the whole axon and dendrite are being stimulated. Compartmentalized activity has a strong role in sensory and interneuron processing, as discussed in PMID: 22722842, 34533987, 35687973, 30609235. This is important especially during bends where local strains can locally activate and suppress neuronal activity through stretch-activated and compression activate channels (PMID 34533987), without structurally determined compartment boundaries.

Specific points:

1.end of the first page: ''modulate mechanosensory response.'' This is very general but the references given only refer to TRNs. I suggest to either change to touch response or include reference that underline the general statement, such as proprioception, nociception etc....

2. it is not really clear what the authors mean by this sentence: ''this may explain why behavior-related neural

activity patterns are seen across the brain in mice, fly and worms, including in nominally sensory areas''

3. The authors should discuss that rig-3 is not exclusively expressed in AVA, which could have consequences on their results. AVA-specific expression of ChR can be found in 10.1038/s41592-023-01836-9

4. Authors need to explain what GtACR2 is? 

5. The notion that the authors claim that they not see a lite-1 dependent blue light activity is surprising and not expected from the literature. E.g. lite-1 has a strong effect in TRNs as well (e.g., doi: 10.1039/C6LC01165A). Are AML496 and 499 the only strains with wt lite-1 when blue light was used? the cleanest results would be to repeat these experiments in a lite-1 bg. Alternatively, the authors should discuss how the presence of lite-1 might influence their results.

6. It is not clear which part of the worm has been illuminated during the turn? The whole worm or only the anterior part?

7. It would significantly accelerate understanding if the authors would summarise in one sentence in the first paragraph in the results section what the experiment was, instead of only citing their previous paper.

Fig. 1: Why is the reversal probability to optogenetic stimulation so low? Wouldn't one expect a higher probability? Is the light intensity low or Chrimson not efficient to cause ChR2 levels of responses?

Supp Fig. 1: mec-4p:chrimson, promotor italics

Supp. Fig. 2: N =1,631 and 10,94 stim events; a number seems missing.

Supp Fig 3: In contrast to all other graphs, the min/max values are 0/1. Why? I suggest to be consistent, unless there is a strog reason.

Reviewer #3: Kumar et al investigate how animals integrate their responses to sensory stimuli with their current behavior. As a paradigm they study processing of information from mechanosensory neurons in C. elegans. C. elegans navigate their environment by alternating between persistent forward and reversals and turns. Aversive stimuli, such as anterior touch, evoke reversals during which the animal backs away from the stimulus, then turns sharply. This behavioral sequence usually leads animals to locomote away from the aversive stimulus. However, for this escape behavior to be successful, animals need to suppress recurrent reversals after the turn to allow them to move away from the noxious cue.

The authors begin by showing that stimulating mechanosensory neurons during a turn is significantly less likely to evoke a reversal than equivalent stimulation when animals are moving forward. They perform this experiments using sophisticated closed loop optogenetic stimulation of gentle touch mechanosensors. Previous work identified a set of neurons that promote reversal behavior that include AIZ, RIM, AIB, AVE and AVA. Most of these neurons are one or two synapses downstream of the anterior touch neurons. The authors use the same strategy of closed loop optogenetics to stimulate each of these neurons in turn while animals are moving forward or turning. Again, they find that stimulating these neurons in turning animals is less likely to evoke a reversal than stimulating them while animals are moving forward. An exception is the AVA neurons, leading the authors to infer that AIZ, RIM, AVD and AVE neurons are at or upstream of the junction where turning circuits inhibit reversal behavior, whereas AVA is downstream They next show that inhibiting neurons previously implicated in turning behavior, namely RIV, SMB and SAA, prolongs the duration of spontaneous reversals. Finally, they show that optogenetically stimulating mechanosensory neurons when animals are moving forward or turning evokes similar rates of reversals if RIV, SAA and SMB neurons are inhibited during turns.This then suggests a model in which neurons that are active during turning (RIV, SMB and SAA) inhibit neurons that promote reversal (AIZ, RIM, AIB, AVE).

The optogenetic approach used by the authors is powerful, and the data they provide supports their circuitry model in which neurons that promote turning inhibit neurons that promote reversal behavior. The usual caveats apply that optogenetic stimulation may not accurately reproduce physiological circuit activity, but the authors have sought to control this as much as possible. The logical next step is to elucidate the molecular mechanisms mediating inhibition of reversal neurons by turning ones. The authors outline one such potential mechanism, mediated by inhibitory acetylcholine receptors, in the discussion, and cite complementary work recently submitted to Biorxiv by Huo and colleagues (citation 48) that provide data supporting this mechanism. 

In summary, the experiments described are well-designed, and the manuscript is clearly written, concise, and easy to understand. I support publication. 

Reviewer #4: "Inhibitory feedback from the motor circuit gates mechanosensory processing in C. elegans" by Kumar et al. uses their previously published method to activate or inhibit particular neurons in worms triggered by particular behaviors. Their previous paper had shown, among other things, that worms' sensitivity to optogenetic activation of the touch neurons depended on their behavioral state. If they were in the process of turning, their probability of reversing in response to the simulated touch was reduced. In the current paper they reproduce this finding with more data and show that isolated turns (initiated during forward locomotion) and escape turns (turns following a reversal) both 'gate' the response to touch neuron stimulation.

The main new findings result from experiments stimulating and inhibiting specific interneurons during the same behavior to identify which are involved in the gating. The authors find that stimulating specific interneurons (AIZ, RIM, AIB, AVE, and AVA) can cause reversals and in all cases except AVA, the probability of an evoked reversal was lower when the worms were turning. They further find that inhibition of turning-associated neurons (RIV, SMB, and SAA simultaneously) increased the length of spontaneous reversals. Finally, they show that inhibiting these neurons rescues the decrease in reversal probability upon optogenetic activation of the touch neurons during reversals. I.e. worms with inhibited RIV, SMB, and SAA reverse at the same rate whether they are turning or going forwards when their touch neurons are activated.

Together these results provide insight into the neural circuits involved in their previous observation of behaviour-modulated sensory responses. The paper is straightforward, well-written, and clear and makes a useful further contribution on top of the original paper.

I have only two minor comments:

-Was expression in the appropriate neurons confirmed for all strains? Especially for the new strains, it would be good to include some supplementary images showing the expression pattern.

-some gene names are not italicized such as lgc-47, acc-1, and acr-3 on page 9.

---

## [Editor Report · Decision Letter 2]

27 Jul 2023

Dear Dr Leifer,

Thank you for the submission of your revised Update Article "Inhibitory feedback from the motor circuit gates mechanosensory processing in Caenorhabditis elegans" for publication in PLOS Biology. Your manuscript has been assessed by the PLOS Biology editorial staff and by the Academic Editor, and we are satisfied by the changes made in response to the reviewers and our previous editorial requests. Therefore, on behalf of my colleagues and the Academic Editor, Piali Sengupta, I am pleased to say that we can in principle accept your manuscript for publication, provided you address any remaining formatting and reporting issues. These will be detailed in an email you should receive within 2-3 business days from our colleagues in the journal operations team; no action is required from you until then. Please note that we will not be able to formally accept your manuscript and schedule it for publication until you have completed any requested changes.

***IMPORTANT REQUEST***: Thank you for expanding your response to Reviewer 2's comment about the presentation of the data, after I contacted you last week, over email. In our editorial system, I updated the Response to Reviewers file and the 'track changes' version of your manuscript, with the new files you provided me. These were the versions that our Academic Editor assessed. However, I was not able to update the 'clean' version of your manuscript, to incorporate the changes you made. Therefore: 

>>Please update the manuscript file in our system to reflect the most recent version, that responds to R2's comments in a bit more detail, including by expanding the description of the statistical analyses, and adding the Supplementary Table S3 that reports the p-values related to each figure presented in the manuscript. 

>>Please also make sure to upload your Table S3 in our system. 

These changes will be required before publication. 

PRESS

We frequently collaborate with press offices. If your institution or institutions have a press office, please notify them about your upcoming paper at this point, to enable them to help maximize its impact. If the press office is planning to promote your findings, we would be grateful if they could coordinate with biologypress@plos.org. If you have previously opted in to the early version process, we ask that you notify us immediately of any press plans so that we may opt out on your behalf.

Sincerely, 

Luke

Lucas Smith, Ph.D.

Senior Editor

PLOS Biology

lsmith@plos.org